# Quantum fidelity susceptibility in excited state quantum phase transitions: application to the bending spectra of nonrigid molecules.

J. Khalouf-Rivera[1], M. Carvajal[1,2], F. Pérez-Bernal[1,2*]

**1** Depto. de Ciencias Integradas y Centro de Estudios Avanzados en Física, Matemáticas y Computación, Universidad de Huelva, Huelva 21071, SPAIN
**2** Instituto Carlos I de Física Teórica y Computacional, Universidad de Granada, Granada 18071, SPAIN
* curropb@uhu.es

March 8, 2021

## Abstract

We characterize excited state quantum phase transitions in the two dimensional limit of the vibron model with the quantum fidelity susceptibility, comparing the obtained results with the information provided by the participation ratio. As an application, we perform fits using a four-body algebraic Hamiltonian to bending vibrational data for several molecular species and, using the optimized eigenvalues and eigenstates, we locate the eigenstate closest to the barrier to linearity and determine the linear or bent character of the different overtones.

# 1 Introduction

The study of bending vibrational degrees of freedom has been fostered due to their two-dimensional nature and the existence of two well-defined physical limits –linear and bent configurations–, together with intermediate configurations –quasilinear species–, characterized by a large amplitude motion that makes them rich in spectroscopic signatures [1]. Positive or non-monotonous anaharmonicities, the latter associated with the occurrence of the Dixon dip in the Birge-Sponer plot for nonrigid molecules [2], and anomalous rotational spectra due to the mixing of linear and bent characters in the wave functions of states straddling in the propinquity of the barrier to linearity [3, 4] are the most salient spectroscopic features that can be found in the spectra of quasilinear species.

Significant advances and developments in spectroscopic methods have made possible the experimental access to high bending overtones for several molecular species. In this way, it has been possible to have access to experimental spectroscopic information that allows for the study of systems at energies around the barrier to linearity [5, 6]. The results obtained for water [7] and NCNCS [8–10] are of particular relevance.

In recent times, the concept of quantum monodromy, initially introduced by Child [11], has greatly helped in the assignment of states in systems where the complexity of wave functions, due to the proximity of the states to the barrier to linearity, hampered a correct state labeling [5–8, 12]. This is a concept borrowed from classical mechanics that relies on the topological singularity happening once the system energy is large enough to probe local saddle points or maxima that prevent the definition of global action angle variables [13].

The theoretical modeling of bending vibrations in nonrigid molecular species requires special tools, as the large amplitude vibrational degree of freedom strongly couples vibrational and rotational degrees of freedom. A pioneering work in this field is the Hougen-Bunker-Johns bender Hamiltonian [14]. This work was later extended to the semirigid bender Hamiltonian [15] and the general semirigid bender Hamiltonian [16]. The MORBID model [17], based on the above mentioned developments, is currently a standard method for the analysis of nonrigid molecular spectra, where the simultaneous consideration of rotational and vibrational degrees of freedom is required for the modeling of experimental term values and the assignment of quantum labels.

The algebraic approach and, in particular, the vibron model is an alternative to the traditional integro-differential approach for the modeling of molecular spectra. This model is based upon symmetry considerations and relies heavily in the properties of Lie algebras [18]. The vibron model (VM) belongs to a family of models that assign a $U(n + 1)$ algebra as a dynamical or spectrum generating algebra for an $n$-dimensional problem [19]. Similar models have been successfully applied to the modeling of the structure of hadrons [20, 21] and nuclei [22–24]. In the original vibron model formalism, introduced by Iachello, rovibrational excitations of diatomic molecular species are treated as collective bosonic excitations [25], and the dynamical algebra is $U(3 + 1) = U(4)$, due to the vector nature of the relevant degrees of freedom [24, 26]. The two-dimensional nature of bending vibrations and the need to simplify the vibron model formalism to efficiently deal with polyatomic systems, naturally drove to the formulation of the two-dimensional limit of the vibron model (2DVM) [27, 28]. The 2DVM defines a formalism that is able to model the linear and bent limiting cases of the bending degree of freedom, as well as the large amplitude modes that characterize intermediate situations [29–32]. An extension to four-body operators of the algebraic Hamiltonian, used

in the present work, has been recently published [33]. The 2DVM has also been used for the modeling of coupled benders [27, 34–36], stretch-bend interations [37–40], and the transition state in isomerization reactions [41].

In recent years, considerable attention has been paid to the occurrence of quantum phase transitions (QPTs) in many different physical systems [42, 43]. Such transitions occur at zero temperature and are due to quantum fluctuations, differing in this way from the usual thermal phase transitions. These transitions are also known as ground state quantum phase transitions due to the abrupt modification experienced by the system ground state wave function once a given parameter in the Hamiltonian (control parameter) goes through a critical value. The work on ground state QPTs in algebraic models can be traced back to the seminal articles by Gilmore *et al.* [44–46] where such transitions were studied for nuclei. These transitions were also called shape phase transitions as each phase corresponds with different geometric configurations of the system's ground state. The study of QPTs in mesoscopic systems is a very active research line [47–50] and a general classification of QPTs in algebraic models can be found in Ref. [51]. In the 2DVM case, the ground state QPT takes place between the linear and bent limiting cases, with a second order phase transition that occurs for nonrigid configurations [29, 30]. A full ground state QPT analysis was performed in [28] and a study of corrections beyond the mean field approach was published in [52]. As this is the simplest two-level algebraic model with a nontrivial angular momentum, it has been chosen in many cases as a test model for QPT studies [53–57].

The study of QPTs was later extended beyond the ground state with the concept of excited state quantum phase transitions (ESQPTs) [58–60]. Such transitions, often associated with a ground state QPT, involve the non-analiticity of the energy level density and level flow for critical values of the energy [61, 62]. For a system with $n$ effective degrees of freedom, the order of the derivative of the level density that is non-analytic is $n-1$ [63–67]. In most cases, ESQPTs can be associated with the existence of an unstable stationary point or a similar singularity in the potential obtained in the classical limit of the system. The non-analiticity is fully realized only in the system large size limit. However, ESQPT precursors can be easily identified for finite systems. Hence, in an ESQPT there exists a borderline of critical energy values, that marks the occurrence of a high level density of states in a certain range of the control parameter or parameters. This line, called separatrix, separates the different ESQPT phases. States belonging to one of the phases have properties akin to the states of the dynamical symmetry associated to the phase in question. As we discuss in this work, it is often cumbersome to assign a given excited state to a phase or to ascertain its position relative to the separatrix. This is particularly complex for systems with several control parameters and a complex phase diagram. ESQPTs have been studied in different quantum many-body systems: the single [61] and coupled [68] Lipkin-Meshkov-Glick models, the Gaudin model [69], the Tavis-Cummings and Dicke models [62, 70], the interacting boson model [49], the kicked-top model [71], periodic lattice models [72, 73], or spinor Bose-Einstein condensates [74, 75]. It has been paid special heed to the influence of ESQPTs on the dynamics of quantum systems [76–86] and to link ESQPT and thermodynamic transitions [87, 88]. For a recent review on the ESQPT subject, with a complete reference list, see Ref. [67].

The 2DVM presents an ESQPT, associated with a second order ground state QPT, that can be explained from the influence on excited states that have enough excitation energy to straddle the barrier to linearity. There is a clear connection between the ESQPT phenomenon and quantum monodromy [28], something that can be generalized to systems other than molecules [89–91]. In fact, the possibility of accessing highly-excited bending levels exper-

imentally makes molecular spectroscopy an optimal playground to detect ESQPTs precursors in experimental spectra [31, 33, 36, 41]. Other systems where ESQPT signatures have been experimentally recorded are superconducting microwave billiards [72] and spinor Bose-Einstein condensates [92].

As mentioned above, states lying at different sides of the separatrix can be ascribed to one or the other of the existing limiting physical situations, or dynamical symmetries. In the 2DVM case, as explained in Sect. 2, states can have a $U(2)$ –linear– or $SO(3)$ –bent– character. However, as one gets further from the limiting cases and closer to the critical energy, it gets cumbersome to assign states to a given phase, due to the strong mixing in the wave function [33]. This explains by the known fact that the definition of order parameters for ESQPTs is not an easy task, in contrast with the situation for ground state QPTs [60].

Recently, a quantity called participation ratio [93] (also known as inverse participation ratio [94] or number of principal components [95]), akin to the Shannon entropy, has been used to quantify the degree of localization of states when expressed in the bases for the different dynamical symmetries. For systems with $U(n + 1)$ dynamical algebra and a second order ground state QPT of the type $U(n)-SO(n+1)$, it has been shown that the participation ratio allows to reveal the location of the ESQPT critical energy due to the enhanced localization of eigenstates with energies close to the critical energy value if they are expressed in the $U(n)$ basis, [79–81]. This fact has been later confirmed, using the 2DVM, in the study of the bending vibrational spectrum of molecular species with large amplitude bending degrees of freedom [33] and in the HCN-HNC isomerization transition state [41]. As explained in Sect. 2, the participation ratio does not allow in all cases for an unambiguous assignment of a linear or bent character to a given state. The large mixing that occurs once the system is far enough from the dynamical symmetry limits hinders this assignment, a fact that can be explained using the quasidynamical symmetry concept [96].

Therefore, it is important to find a quantity other than the participation ratio that allows for the unambiguous assignment of 2DVM excited states around an ESQPT to one of the implied phases. In recent times, quantum-information-derived quantities have been successfully employed to characterize ground state QPTs as they offer an approach that does not rely on the identification of an order parameter and its corresponding symmetry-breaking pattern (see [97–99] and references therein). Inspired by these works, we have found that we can unambiguously assign excited states into ESQPT phases using quantum fidelity or quantum fidelity susceptibility. Quantum fidelity is a concept that arises in quantum information theory and it involves the overlap of wave functions [100]. This quantity has been successfully applied to the study of ground state quantum phase transitions and critical phenomena [101]; for a review see [98]. A derived quantity that has been used for the characterization of QPTs is the quantum fidelity susceptibility (QFS), the second derivative of the fidelity and the leading order term in the series expansion of the fidelity [98, 102, 103]. In the present work we extend the calculation of quantum fidelity and QFS to 2DVM excited states. We obtain an unambiguous assignment of such states to one of the possible ESQPT phases, as we can locate the state position relative to the separatrix between ESQPT phases. We apply the formalism to a recently presented four-body 2DVM Hamiltonian [33], and we assign a linear or bent character to the excited states of linear and non-rigid molecular bending vibrations.

The present work is structured as follows. We provide a brief introduction to the 2DVM in Sect. 2, presenting a simple model Hamiltonian and the two possible dynamical symmetries. In Sect. 3, we discuss the participation ratio and the quantum fidelity results for the model Hamiltonian. In Sect. 4, we introduce the four-body Hamiltonian used to fit to experimental

data and we apply the formalism to the $Si_2C$ molecule, a well-known example of nonrigid molecule [6]. Finally, in an abridged form, we also show results for selected bending degrees of freedom for other five molecular species. In Sect. 5, we include a summary of the work and some concluding remarks.

# 2  The two-dimensional limit of the vibron model

Due to the two-dimensional nature of bending vibrations, its algebraic modeling implies the treatment of vibrational quanta as collective bosonic excitations using the $U(3)$ Lie algebra as a dynamical algebra [27, 28].

Following the algebraic formalism [24, 26], one should consider the possible dynamical symmetries that are subalgebra chains starting in the dynamical algebra and ending in the system's symmetry algebra. The conservation of angular momentum in a bending mode (vibrational angular momentum) implies that the symmetry algebra in this case is $SO(2)$. There exists two possible chains that start in $U(3)$ and end in $SO(2)$ [1]

$$
\begin{array}{cccccl}
U(3) & \supset & U(2) & \supset & SO(2) & \text{Chain (I)} \\
N & & n & & \ell & \\
U(3) & \supset & SO(3) & \supset & SO(2) & \text{Chain (II)} \\
N & & \omega & & \ell &
\end{array}
\tag{1}
$$

Each dynamical symmetry provides a set of quantum labels and a basis to treat the problem of bending vibrational spectra and it is associated with a limiting physical case. Chain (I) is known as the cylindrical oscillator chain and it can be mapped with the bending vibrations of a linear molecule. Its associated basis is a truncated 2D harmonic oscillator basis, with quantum labels $\{|[N]\,n^\ell\rangle\}$. The quantum label $N$ identifies the totally symmetric representation of $U(3)$ and determines the size of the system Hilbert's space; the $n$ and $\ell$ labels are the number of quanta of excitation in the 2D harmonic oscillator and the vibrational angular momentum, respectively. Chain (II) is known as the displaced oscillator chain and it can be mapped to the limiting physical case of a bent molecule. The associated basis is expressed as $\{|[N]\,\omega\,\ell\rangle\}$ where $N$ has the same interpretation than in the previous case, $\omega$ can be connected with $\nu_b$, the number of quanta of excitation in the anharmonic displaced oscillator: $\nu_b = (N - \omega)/2$. Finally, $\ell$, is the projection of the molecular angular momentum on the figure axis. In some cases this quantity is expressed using the usual notation for symmetric tops $\ell = K$.

The branching rules define the allowed range of values for the quantum numbers in Eq. (1).

$$
\begin{array}{llll}
n = 0, 1, 2, \ldots N & \ell = \pm n, \pm(n-2), \ldots, \pm 1 (\text{or } 0) & \text{Chain (I)} \\
\omega = N, N-2, \ldots 1 (\text{or } 0) & \ell = \pm \omega, \pm(\omega - 1), \ldots, \pm 1, 0 & \text{Chain (II)}
\end{array}
\tag{2}
$$

The calculations can be performed in any of the two basis, and the selection of one or the other is often determined by the nature of the system under study. The transformation brackets between both bases can be analytically derived [28, 104].

In the algebraic approach, the Hamiltonian and any other operator of interest is expressed as a function of Casimir or invariant operators of the subalgebras in the different dynamical symmetries. A specially convenient and simple model Hamiltonian can be built using two

---

[1]A third chain, $U(3) \supset \overline{SO}(3) \supset SO(2)$ can be defined but it has the same physical interpretation than the $U(3) \supset SO(3) \supset SO(2)$ and it does not add new features to the model [28].

operators: the number operator, $\hat{n}$, and the pairing operator, $\hat{P} = N(N+1) - \hat{W}^2$. The first one is the first order Casimir of the $U(2)$ subalgebra while the second one is built with $\hat{W}^2$, the second order Casimir operator of the $SO(3)$ subalgebra. The model Hamiltonian,

$$\hat{\mathcal{H}}(\xi) = \varepsilon \left[ (1 - \xi)\hat{n} + \frac{\xi}{N - 1}\hat{P} \right] \;, \tag{3}$$

depends on two parameters, the system control parameter, $\xi \in [0, 1]$, and the energy scale, $\varepsilon$. Hereafter we fix the energy scale to $\varepsilon = 1$ and the calculated energies are dimensionless quantities. The pairing operator is a two-body operator, while the number operator is a one-body operator; therefore the two-body part is normalized by the system size to make the Hamiltonian intensive and allow for the calculation of results in the thermodynamic or large size –large $N$– limit. The study of the eigenvalues of Hamiltonian (3) and its classical limit determines that there exists a ground state QPT of second order with a critical control parameter value $\xi_c = 0.2$ [28]. For control parameter values $\xi \leq \xi_c$, the system is said to be in the $U(2)$ or symmetrical phase, which in the molecular case can be mapped to a linear configuration. If $\xi > \xi_c$, then the system is in a $SO(3)$ or broken symmetry phase, known as a bent –or semirigid– configuration in the case of vibrational bending.

The conservation of vibrational angular momentum implies that Hamiltonian (3) is block diagonal in the quantum label $\ell$. The nonzero matrix elements in the cylindrical oscillator basis are

$$\langle [N]n_2^\ell | \hat{\mathcal{H}}(\xi) | [N]n_1^\ell \rangle = \left[ (1 - \xi)n_1 + \frac{\xi \left\{ N(N+1) - (N - n_1)(n_1 + 2) - (N - n_1 + 1)n_1 - \ell^2 \right\}}{N - 1} \right] \delta_{n_2, n_1}$$

$$+ \frac{\xi}{N - 1}\sqrt{(N - n_1 + 2)(N - n_1 + 1)(n_1 + \ell)(n_1 - \ell)} \, \delta_{n_2, n_1 - 2} \tag{4}$$

$$+ \frac{\xi}{N - 1}\sqrt{(N - n_1)(N - n_1 - 1)(n_1 + \ell + 2)(n_1 - \ell + 2)} \, \delta_{n_2, n_1 + 2} \;.$$

If $\xi = 0$, the Hamiltonian is diagonal in Chain (I) basis and the spectrum is harmonic; while in the $\xi = 1$ case the Hamiltonian is diagonal in Chain (II) basis and the spectrum is anharmonic, with degenerate rotational bands. This can be clearly seen in the correlation energy diagram depicted in the left panel of Fig. 1, where the excitation energy is plot as a function of the control parameter $\xi$ for $N = 10$ using full red lines for levels with even angular momentum values and dashed blue lines for odd angular momentum levels. This plot allows to track level paths from one dynamical symmetry to the other; from a 2D harmonic spectrum on the left side ($\xi = 0$) to the anharmonic oscillator spectrum in the right side ($\xi = 1$). We have included the quantum labels $n^\ell$ in the $U(2)$ limiting case and the number of quanta of excitation $\nu_b$ associated to the $\omega$ quantum label in the $SO(3)$ case. In the latter case, once $\xi = 1$, values with different vibrational angular momentum form a degenerate rotational band. In between these two cases, the model Hamiltonian (3), despite its simplicity, is able to reproduce spectra with positive anharmonicity, associated with flat potentials for control parameter values less than $\xi_c$. It can also reproduce the spectroscopic signatures of nonrigid molecular species: the Dixon dip and the change from a linear to a quadratic dependence of the energy with vibrational angular momentum that characterizes quantum monodromy [28, 29, 31–33].

We also plot the correlation energy diagram for Hamiltonian (3) in the right panel of Fig. 1, this time for a larger system size ($N = 80$) and including only $\ell = 0$ (full red lines)

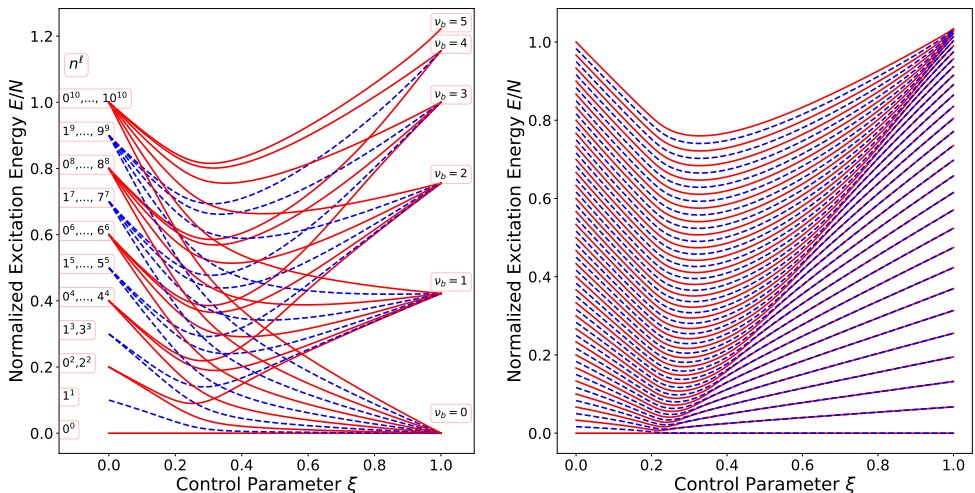

Figure 1: Normalized excitation energy of the 2DVM Hamiltonian (3) as a function of the control parameter $\xi$. Left panel: system size is $N = 10$ and all angular momenta states are shown ($\ell = 0 \ldots 10$) using full red (dashed blue) lines for even (odd) angular momentum values. Right panel: system size is $N = 80$ and states with $\ell = 0$ ($\ell = 1$) states are depicted using full red (dashed blue) lines.

and $\ell = 1$ (dashed blue lines) states. In this case, it is evinced a line marked by a locally high density of energy levels, starting at $\xi = \xi_c = 0.2$ and zero excitation energy. This line marks the ESQPT critical energy and the boundary between the two ESQPT phases. It can also be easily appreciated that in the $SO(3)$ phase, below the line, states with different vibrational angular momenta are degenerate and this degeneracy is broken in the $U(2)$, for states above the line.

# 3 Quantum Fidelity Susceptibility in the 2DVM

A convenient tool for the characterization of wave functions in the phases defined by an ESQPT is the participation ratio (PR). This quantity provides the degree of localization of a state in the available bases [93] –it is also known as inverse participation ratio [94] or number of principal components [95]. For a quantum state $|\Psi\rangle$, expressed in a given basis $\{|\phi_i\rangle\}_{i=1,\ldots,\dim}$ as $|\Psi\rangle = \sum\limits_{i=1}^{\dim} c_i |\phi_i\rangle$, the PR is defined as

$$PR[\Psi] = \frac{1}{\sum\limits_{i=1}^{dim} |c_i|^4} . \tag{5}$$

Note that the minimum value of the PR for a given state is one, and this means that the state under scrutiny belongs to the basis. On the other hand, the maximum value is equal

to the basis dimension, $dim$, in the –nonrealistic– case of a state with equal and non zero $c_i$ coefficients ($c_i = 1/\sqrt{dim}$).

In algebraic models with an ESQPT associated with $U(n) - SO(n+1)$ dynamical symmetries,i t has been found that state(s) close to the critical energy of the ESQPT display a high localization in one of dyanamical symmetry basis [79–81]. The main application of this quantity in the 2DVM stems from the high localization of the $\ell = 0$ states that lie closer to the barrier to linearity –critical energy of the ESQPT– when expressed in the the $U(2)$ basis (1). More precisely, such state, or states, have a dominant component for the basis state $|[N]n = 0^{\ell=0}\rangle$ [33]. This effect is blurred for increasing values of the vibrational angular momentum, $\ell$. This is an effect that can be explained by the centrifugal barrier precluding the wave function from exploring the barrier to linearity critical point. The PR has also proved useful in the caracterization of different type of ESQPT, associated with the $\ell = 0$ transition states in isomerization reactions, where the $|[N]n = N^{\ell=0}\rangle$ component has the highest weight in the Chain (I) basis [41].

In the 2DVM, cases with an ESQPT –i.e. with quantum monodromy–, notwithstanding the critical energy of the ESQPT is well determined from the PR values for eigenstates in the $U(2)$ basis, the comparison of the PR values obtained for the $U(2)$ and the $SO(3)$ bases does not allow for a clear assignment of a given eigenstate to a linear or bent ESQPT phase. This is specially relevant for systems with a low barrier to linearity, and for states that lie far from both limiting physical cases, the $U(2)$ and $SO(3)$ dynamical symmetries. This is in good accordance with the quasidynamical symmetry concept [96], that explains the high degree of mixing expected as one gets further from the dynamical symmetries, even for states retaining most of the characteristic features of a dynamical symmetry. Therefore, in such cases, the direct comparison of the PR values for the $U(2)$ and $SO(3)$ bases does not allow an unambiguous assignment of the eigenstate to a linear or bent character.

Thus, we have looked for a basis-independent quantity that could achieve an unambiguous assignment of a system excited states to one of the existing phases in a precise manner. This is specially relevant once we move from the simple model Hamiltonian (3) to more complex Hamiltonians that include higher order operators.

Our proposal is to extend the QFS to 2DVM excited eigenstates, obtaining in this way a sensitive probe, able to locate a given eigenstate position with respect to the separatrix line between ESQPT phases. It therefore allows to assign excited states to a $U(2)$ or $SO(3)$ ESQPT phase in a basis-independent way. We proceed to define QFS and its application to the 2DVM.

The definition of quantum fidelity, a quantity introduced in quantum information theory [100], for a system with a single control parameter, $\lambda$, is

$$F(\lambda, \delta\lambda) = |\langle \psi_0(\lambda)|\psi_0(\lambda + \delta\lambda)\rangle| \ . \tag{6}$$

This quantitiy provides a measure of the similarity between ground quantum states obtained for control parameters values $\lambda$ and $\lambda + \delta\lambda$. Despite its apparent simplicity, this quantity efficiently grasps the sudden change experienced by the ground state wave function once the control parameter is varied across its critical value and, since the seminal work of Zanardi [101], it has been used to characterize QPTs in different systems [98,102]. Another magnitude often used to identify QPTs is the QFS [98,102], which is maximum when the parameter $\lambda$ goes through a critical value

$$\chi_F(\lambda) = -\frac{\partial^2 F(\lambda, \delta\lambda)}{\partial(\delta\lambda)^2} = \lim_{\delta\lambda \to 0} \frac{-2\ln F(\lambda, \delta\lambda)}{(\delta\lambda)^2} \ . \tag{7}$$

Using perturbation theory, the QFS can be expressed in the so called summation form [98]

$$\chi_F(\lambda) = \sum_{i \neq 0}^{dim} \frac{\left| \langle \psi_i(\lambda) | \hat{H}^I | \psi_0(\lambda) \rangle \right|^2}{[E_i(\lambda) - E_0(\lambda)]^2} \ , \tag{8}$$

where $\hat{H}^I$ is the interaction Hamiltonian and the total Hamiltonian can be written as $\hat{H}(\lambda) = \hat{H}^0 + \lambda H^I$; $|\psi_i(\lambda)\rangle$ is the $i$-th eigenvector of Hamiltonian $\hat{H}(\lambda)$ and $E_i(\lambda)$ is its eigenvalue. An important advantage of the QFS, expressed in this form, is that it is independent of the $\delta\lambda$ value.

The QFS has been used in the characterization of ground state quantum phase transitions and their universality in relevant many-body quantum systems, e.g. the 1D Hubbard model [102, 105], the Kitaev honeycomb model [106], the 1D asymmetric Hubbard model [103], the Lipkin-Meshkov-Glick model [107–109], the two-dimensional transverse-field Ising and XXZ models [110], the Rabi model [111], Gaussian random ensambles [112], or 1D lattice models [113–116].

In the present work, we extend the concept of QFS beyond the ground state, to the realm of excited states, and we use this magnitude as a probe to locate excited states in the 2DVM with respect to the separatrix line between different ESQPT phases. We will apply this to the results obtained in the fit of Hamiltonian (13) to several molecular species obtaining an unambiguous assignment of the excited states to a given basis.

Our proposal is to introduce a control parameter $\lambda$ and split the algebraic 2DVM spectroscopic Hamiltonian into three different terms: a first one, $\hat{H}_I$, that encompasses all operators diagonal in the $U(2)$ basis and its associated spectroscopic parameters; a second one, $\hat{H}_{II}$, including terms diagonal in the $SO(3)$ basis; and a third one, $\hat{H}_{I-II}$, containing operators and the corresponding spectroscopic parameters diagonal in both bases

$$\hat{H}(\lambda) = (1 - \lambda)\hat{H}_I + (1 + \lambda)\hat{H}_{II} + \hat{H}_{I-II} = \hat{H}(\lambda = 0) + \lambda\hat{H}^I \ , \tag{9}$$

$$\hat{H}^I = -\hat{H}_I + \hat{H}_{II} \ . \tag{10}$$

The control parameter $\lambda$ is defined in the range $\lambda \in [-1, 1]$ and the initial Hamiltonian is recovered for $\lambda = 0$. The Hamiltonian $\hat{H}(\lambda = \pm 1)$ is diagonal in the $U(2)/SO(3)$ basis.

We now proceed to define the QFS for the $j-$th eigenstate of Hamiltonian $\hat{H}(\lambda)$ as

$$\chi_F^{(j)}(\lambda) = \sum_{i \neq j}^{dim} \frac{\left| \langle \psi_i(\lambda) | \hat{H}^I | \psi_j(\lambda) \rangle \right|^2}{[E_i(\lambda) - E_j(\lambda)]^2} \ , \tag{11}$$

that is a generalization of Eq. (8) to excited states. As the value of the $\lambda$ control parameter is varied, $\chi_F^{(j)}(\lambda)$ will evidence –even for finite-size systems– a peak whenever a separatrix line associated with an ESQPT is crossed. A similar procedure has been recently published, using QFS of excited states, in the study of the adiabatic and counter-adiabatic driving in ESQPTs [67] and of the onset of quantum chaos in spin chain models [116].

We show as an example the application of Eq. (11) to the excited states of model Hamiltonian (3) for a fixed $\xi > 0.2$. In this case, $\hat{H}_I(\xi) = (1 - \xi)\hat{n}$ and $\hat{H}_{II}(\xi) = \xi/(N - 1)\hat{P}$, and

$$\hat{H}(\lambda) = \hat{\mathcal{H}}(\xi) + \lambda\hat{H}^I(\xi) \ , \tag{12}$$

where $\hat{H}^I(\xi) = -(1-\xi)\,\hat{n} + \left(\frac{\xi}{N-1}\right)\hat{P}$ and the new control parameter is $\lambda$. We show the results obtained for $\ell = 0$ states of the model Hamiltonian with a control parameter value $\xi = 0.6$ and a system size $N = 200$ in Fig. 2. The correlation energy diagram, plotting the normalized excitation energy versus the $\lambda$ control parameter, is shown in the upper panel. The resulting diagram is, as expected, similar to the correlation energy of $\hat{\mathcal{H}}(\xi)$, with a ground state QPT and a line of high density of states that marks the ESQPT separatrix. The energies for $\lambda = 0$ are the energies of our selected model Hamiltonian case. We have highlighted the results obtained for the ground state and the states with normalized excitation energies closer to $0.05, 0.2, 0.4, 0.6$ , and $0.8$, with different colors (orange, light green, purple, pink, cyan, and dark green, respectively). Therefore, instead of going across the ESQPT following a given eigenstate, we have selected a set of states according to their excitation energy values. For each one of them, the ESQPT separatrix is crossed at different $\lambda$ values (see upper panel).

We plot in Fig. 2 center panel the results obtained for the QFS (11) normalized by the system size for the model Hamiltonian excited states as a function of $\lambda$. We use the same color code than in the upper panel to emphasize the results for a selected set of states. It is clear that the QFS for an excited state reaches its maximum value when the state energy straddles the ESQPT critical energy line. Therefore, if the maximum of the QFS for a level occurs for a negative (positive) $\lambda$ value, the level lies below (above) the ESQPT separatrix and we can assign a $SO(3)$ ($U(2)$) character to the excited state. In case the QFS maximum value is obtained for $\lambda = 0$, the system excited state energy coincides with the critical energy and the state is in the separatrix line. In the provided example, the ground state and the states with normalized energy values close to $0.05, 0.2$, and $0.4$ are of bent ($SO(3)$) type and the states with $E/N$ close to $0.6$ and $0.8$ have a linear ($U(2)$) character. The eigenstate $\nu_b = 50$ is the state with an energy closest to the separatrix for $\lambda = 0$.

The lower panel of Fig. 2 shows, for the same selected states and with the same color code, the value of the normalized PR in the $U(2)$ basis as a function of the $\lambda$ control parameter. In the ground state case, there is an abrupt change in the PR value for the $\lambda$ value associated with the ground state quantum phase transition, while excited states show a minimum in the participation ratio for the $\lambda$ control parameter value that makes them cross the ESQPT separatrix, as predicted in Refs. [79–81].

In order to further illustrate the role of the QFS, we plot in the upper panel of Fig. 3 the normalized QFS –blue dashed line, left ordinate axis scale– and PR –red solid line, right ordinate axis scale– as a function of the normalized excitation number $(2\nu_b/N)$ for the model Hamiltonian eigenstates, with $\lambda = 0$, $\xi = 0.6$, and $N = 200$. Both quantities can be used to assess the value of the ESQPT critical energy, indicated by a minimum (maximum) value of the PR (QFS).

The lower panel of Fig. 3 shows $\lambda_{\max}$, the value of the control parameter $\lambda$ for which each eigenstate of the model Hamiltonian with $\xi = 0.6$ and $N = 200$ has a maximum QFS value as a function of the normalized excitation number. The horizontal black dashed line marks the $\lambda = 0$ value and, as previously stated, eigenstates with $\lambda_{\max} < 0$ ($\lambda_{\max} > 0$) can be classified as linear- (bent-like) states.

Therefore, the QFS provides a trustworthy and basis-independent method to locate states with respect to the high level density separatrix lines that characterize ESQPTs. The case (12) is a particularly simple one, but in the next section we show how to use the QFS in a more general case, with an application to the bending wavefunctions obtained from the fit of a Hamiltonian including up to four-body interactions to reported vibrational bending

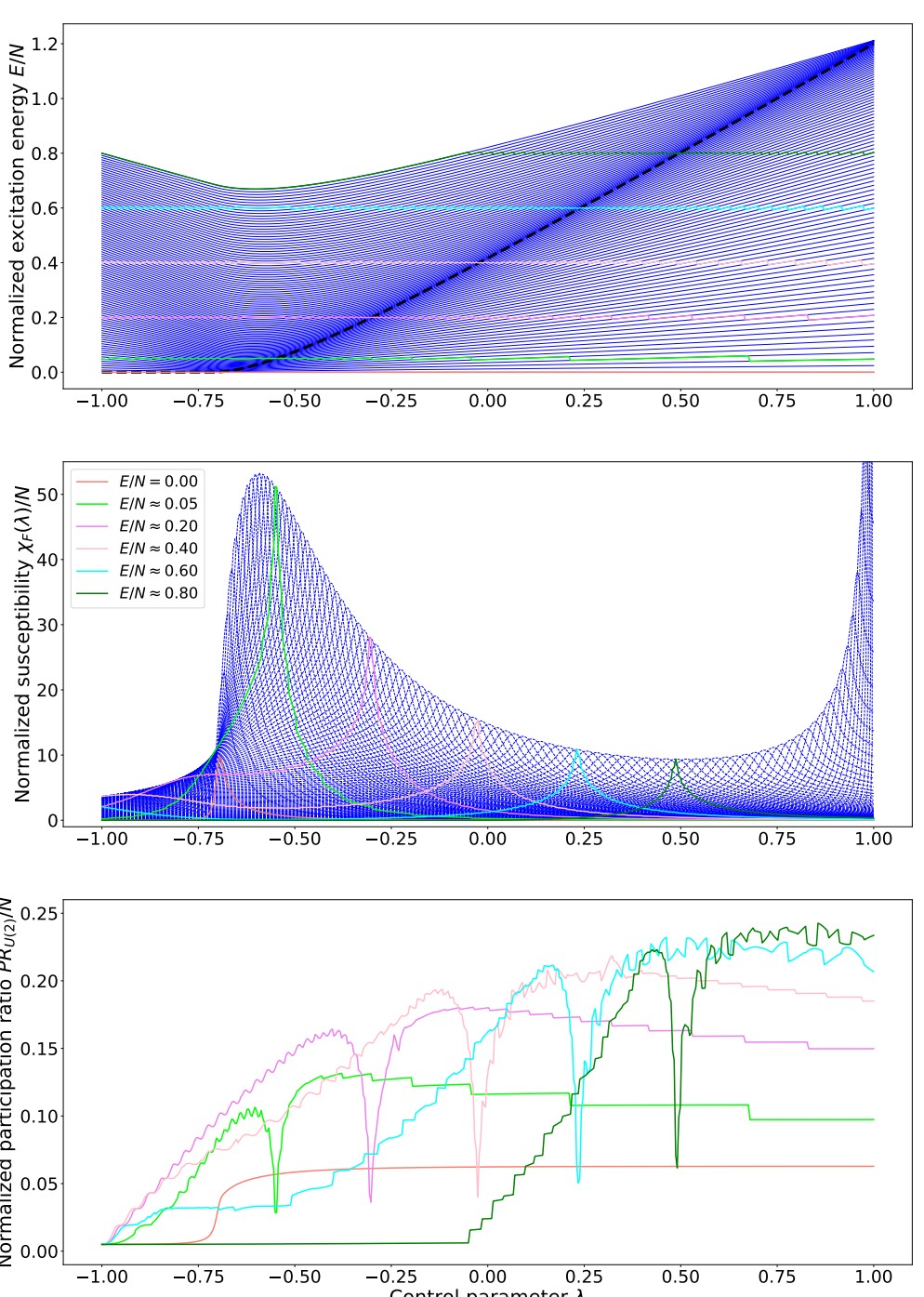

Figure 2: All panels: Results for a 2DVM Hamiltonian (12) with $\xi = 0.6$ and $N = 200$. For the sake of clarity, the ground state and states with normalized excitation energies closer to $0.05, 0.2, 0.4, 0.6$ , and $0.8$ have been highlighted using different colors (orange, light green, purple, pink, cyan, and dark green, respectively). The rest are plotted with blue dashed lines. All quantities are plotted versus the control parameter $\lambda$. Upper panel: Normalized excitation energy of $\ell = 0$ eigenstates of the model Hamiltonian (12). Middle panel: Normalized QFS (11). Lower panel: Normalized PR in the $U(2)$ basis (5) for the set of selected states indicated above.

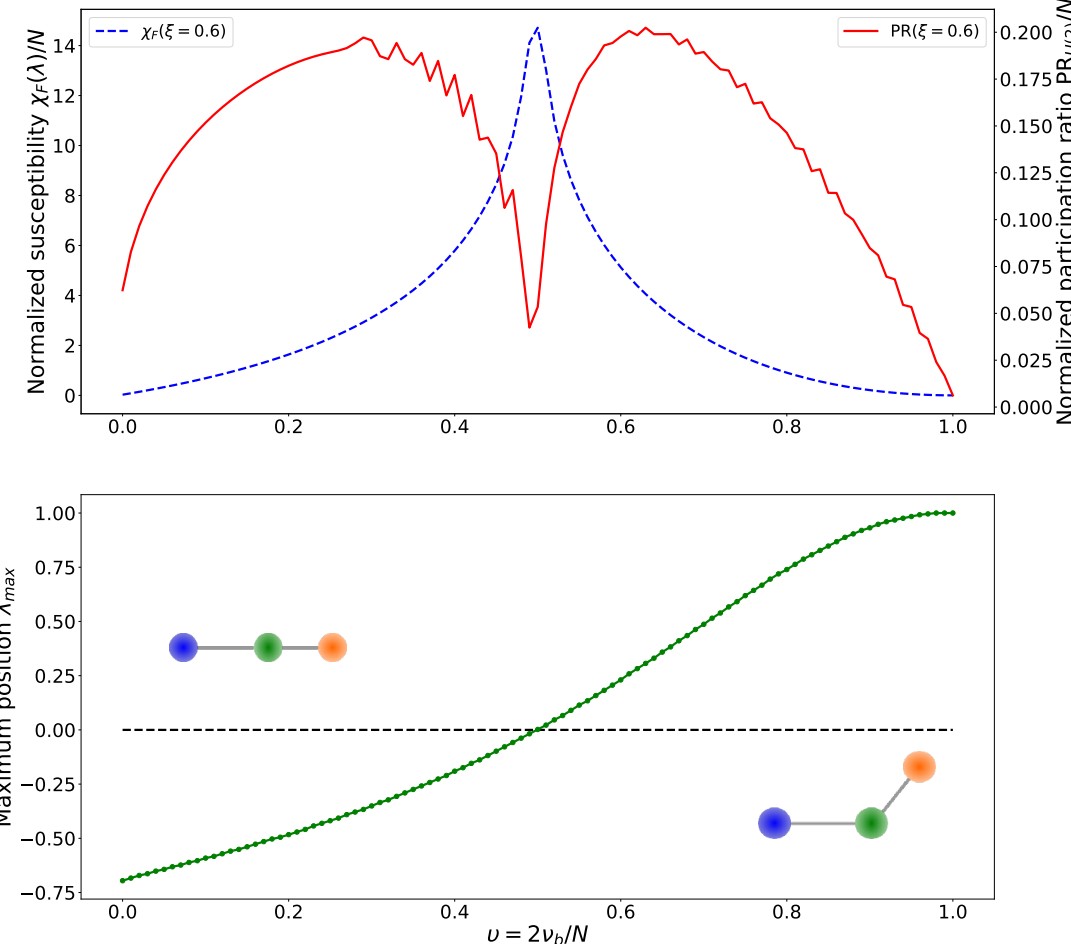

Figure 3: All panels: Results for the 2DVM model Hamiltonian (12) with $\xi = 0.6$ and $N = 200$. Upper panel: PR (red solid line, right ordinate axis scale) and QFS (blue dashed line, left ordinate axis scale) for the $\lambda = 0$ eigenstates. Lower panel: Position of the QFS maxima, $\lambda_{\max}$, for each eigenstate versus the normalized bending quantum number $2\nu_b/N$. The black dashed line marks the $\lambda = 0$ value.

band origins for different molecular species. In fact, the difficulty of clearly assigning levels in cases such as the ones included in the next section has been the original motivation for this research [33].

# 4 Application to molecular bending structure

In order to illustrate how PR and QFS can help in the characterization of bending vibrational excited states we apply the procedure explained in section 3 to reported data for several molecules. We have selected mostly nonrigid species, due to their feature-rich bending spectrum that includes an ESQPT once eigenstates straddle the barrier to linearity. In particular, we show results for bending data for $Si_2C$, NCNCS, HNC, $CH_3NCO$, $^{37}ClCNO$, and OCCCO.

Notwithstanding model Hamiltonian (3) has the basic ingredients to model the limiting linear and bent cases, as well as the rich gamut of intermediate situations, it is too simple to attain experimental accuracy in fits of observed band origins for bending degrees of freedom. Previous fits have been performed, in most cases, using the general one- and two-body algebraic Hamiltonian [29, 31, 32]; adding higher order interactions for specially hard cases, as in the case of the bending vibrational spectrum of water [32]. We have recently presented improved results from a systematic study using the most general Hamiltonian that includes up to four-body interactions [33].

$$\begin{aligned}
\hat{H}_{4b} =& P_{11}\hat{n} \\
&+ P_{21}\hat{n}^2 + P_{22}\hat{\ell}^2 + P_{23}\hat{W}^2 \\
&+ P_{31}\hat{n}^3 + P_{32}\hat{n}\hat{\ell}^2 + P_{33}(\hat{n}\hat{W}^2 + \hat{W}^2\hat{n}) \\
&+ P_{41}\hat{n}^4 + P_{42}\hat{n}^2\hat{\ell}^2 + P_{43}\hat{\ell}^4 + P_{44}\hat{\ell}^2\hat{W}^2 \\
&+ P_{45}(\hat{n}^2\hat{W}^2 + \hat{W}^2\hat{n}^2) + P_{46}\hat{W}^4 + P_{47}(\hat{W}^2\overline{\hat{W}}^2 + \overline{\hat{W}}^2\hat{W}^2)/2 \ .
\end{aligned} \tag{13}$$

The notation for the algebraic spectroscopic parameters, $P_{ij}$, indicates that this is the $j-th$ parameter for $i$-th body interactions. The matrix elements of the different operators in Hamiltonian (13) in the two possible basis –$U(2)$ and $SO(3)$– can be found in Ref. [33], where the authors have recently published a fit to bending data of $Si_2C$, NCNCS, and HNC using Hamiltonian (13), obtaining a very satisfactory agreement with the reported band origins. The interpretation of the PR for the resulting eigenstates [33], is hampered by the new ESQPT features introduced by three- and four-body interactions in Hamiltonian (13) as it was already shown, for a simpler case, in Ref. [117]. This provides further support to the use of a basis-independent alternative quantity as the QFS.

In the present work, we carry out similar fits to a selected bending mode of the $CH_3NCO$, $^{37}ClCNO$, and OCCCO molecules. Though the influence of all spectroscopic parameters in (13) was explored for each case under study, not all of them are needed in the fit and you can find a summary of our results in Tab. 1. For the sake of brevity, we explain it in detail the results for $Si_2C$, whereas the results obtained for the rest of the molecules are more succinctly reported. Some extra details can be found in the figures included in Appendix.

In the Hamiltonian (13) the three- and four-body operators $\hat{n}\hat{W}^2 + \hat{W}^2\hat{n}$, $\hat{n}^2\hat{W}^2 + \hat{W}^2\hat{n}^2$, and $(\hat{W}^2\overline{\hat{W}}^2 + \overline{\hat{W}}^2\hat{W}^2)/2$ are built as symmetrized products of Casimir operators and, therefore, are not diagonal neither in the $U(2)$ nor the $SO(3)$ bases. To take this fact into account,

we extend the definitions (9,10) to include such operators

$$\hat{H}\left(\lambda\right) = \left(1-\lambda\right)\hat{H}_I + \left(1+\lambda\right)\hat{H}_{II} + \left(1-\lambda^2\right)\hat{H}_{\mathrm{mix}} + \hat{H}_{I-II} \,, \tag{14}$$

where $\hat{H}_I$, $\hat{H}_{II}$, and $\hat{H}_{I-II}$ have the same meaning explained in Eq. (9,10), and $\hat{H}_{\mathrm{mix}}$ encompasses those interactions that are diagonal in neither the $U(2)$ nor the $SO(3)$ basis. In this case, and applying first order perturbation theory, the interaction Hamiltonian is $\hat{H}^I = -\hat{H}_I + \hat{H}_{II} - 2\lambda\hat{H}_{\mathrm{mix}}$. Again, the original Hamiltonian is recovered for $\lambda = 0$ and the Hamiltonian $\hat{H}\left(\lambda = \pm 1\right)$ is diagonal in the $U(2)/SO(3)$ basis. Considering this definition, the QFS can be computed using Eq. (11).

## 4.1 Detailed study of the Si$_2$C case

The available data for the large amplitude bending degree of freedom of Si$_2$C [6] were studied using the four-body 2DVM Hamiltonian (13), obtaining already a fit within experimental accuracy considering one- and two-body operators (fitting spectroscopic parameters $P_{11}$, $P_{21}$, $P_{22}$, and $P_{23}$). The number of available observed term values is 37, with $\nu_b$ up to 13 and a maximum vibrational angular momentum $\ell = 3$ [6]. The resulting fit has $rms$=1.48 cm$^{-1}$ [33], of the same order than the reported experimental uncertainty (2.0 cm$^{-1}$) [6]. The values of the total vibron number, $N$, the optimized spectroscopic parameters and their one-sigma uncertainty, and the fit $rms$ are shown in Tab. 1. The interested reader can find a detailed description of the fitting procedure in Ref. [33].

In this case, as all operators are diagonal in either the $U(2)$ or the $SO(3)$ basis (1), the control parameter $\lambda$ dependent Hamiltonian (9) can be written as follows

$$\begin{aligned}
\hat{H}_{\mathrm{Si_2C}}\left(\lambda\right) &= \left(1-\lambda\right)\ \left[P_{11}\hat{n} + P_{21}\hat{n}^2\right]\ +\ \left(1+\lambda\right)\ \left[P_{23}\hat{W}^2\right]\ +\ \left[P_{22}\hat{\ell}^2\right] \\
&= \left(1-\lambda\right)\qquad\quad \hat{H}_I \qquad\quad +\ \left(1+\lambda\right)\qquad \hat{H}_{II} \qquad +\quad \hat{H}_{I-II}
\end{aligned}$$

The optimized Hamiltonian parameter values are obtained for $\lambda = 0$, as $\lambda$ approaches a value of 1 ($-1$) only terms associated with the $SO(3)$ ($U(2)$) dynamical symmetry are nonzero.

We proceed to calculate the QFS as a function of $\lambda$ for the first eleven Si$_2$C bending eigenstates, well beyond the barrier to linearity. The obtained results, for vibrational angular momentum $\ell = 0, 1$, and 2, are depicted in the left column panels of Fig. 4. For each state, the QFS is maximal at a certain $\lambda$ value. This result is completely equivalent to the result presented in Fig. 2 for the model Hamiltonian (3): the maximum QFS value indicates what is the $\lambda$ value for which the state under study crosses the high-density of states ESQPT separatrix line. Therefore, if the maximum occurs for a negative $\lambda$ value, an originally bent state (belonging to the $SO(3)$ or broken symmetry phase) is changing to a linear state (that belongs to the $U(2)$ or symmetric phase) and vice versa for a maximum at a positive $\lambda$ value. In nonrigid molecules, where an ESQPT is expected, the level whose QFS maximum is the closest to $\lambda = 0$ is the bending eigenstate with an energy that is equal to the ESQPT critical energy. This state can be considered the transition state from bent to linear configurations. The states of the left column of Fig. 4 can be labeled attending to their maximum position: from left to right we have included states $\nu_b = 0, 1, ..., 10$ and, in the Si$_2$C case, the transition state is the fifth bending overtone, $\nu_b = 6$.

The right column panels in Fig. 4 show the participation ratio values in the $U(2)$ basis for bending levels $\nu_b = 0, 3, 6$, and 9, and for vibrational angular momentum values $\ell = 0, 1$, and

2. As already mentioned in the discussion of the lower panel of Fig. 2, in the calculation of the participation ratio for the model Hamiltonian, the system ground state is better localized in the $U(2)$ basis before the ground state QPT, and the PR value suddenly increases once the system goes through the critical point. As this sudden change takes place for a negative $\lambda$ value, that implies that the ground state of $Si_2C$ is a bent-like state, as expected. In the excited levels case, the PR is minimal once each wave function gets through the ESQPT separatrix, being $\nu_b = 6$ the most localized state for $\lambda = 0$. We have performed also calculations of the QFS for the optimized $Si_2C$ states ($\lambda = 0$ case) with vibrational angular momentum $\ell = 0, 1$, and 2. The obtained results are shown in Fig. 6 in Appendix A; and they display the same trends than the model Hamiltonian case shown in the upper panel of Fig. 3.

As expected, in both cases, for the PR and the QFS, ESQPT precursors are weaker for higher $\ell$ values (see also Fig. 6 in Appendix A). This is a well-known effect explained by the centrifugal barrier hindering the access of the wavefunction to the maximum in the barrier to linearity [60].

## 4.2 Application to other molecules

The detailed study of the $Si_2C$ case evinces the efficiency of the QFS locating excited states with respect to the ESQPT separatrix. In the present subsection, we extend the study to other molecular species. We include a linear molecule (HNC) and four nonrigid species ($CH_3NCO$, $^{37}ClCNO$, OCCCO, and NCNCS) with different barrier to linearity heights.

As in the $Si_2C$ case, the fits for the HNC and NCNCS are the same than the authors have recently presented in [33], using the 2DVM four-body Hamiltonian (13). In short, the HNC fit included terms up to two-body interactions plus a three-body term, $\hat{n}\hat{\ell}^2$ ($P_{32}$ parameter), to fit 19 experimental data [118] with $N = 40$. The obtained fit precision is very satisfactory, with an $rms = 0.08\,\mathrm{cm}^{-1}$. The modeling of the NCNCS nonrigid bending degree of freedom (CNC bending, $\nu_7$ normal mode) implies the inclusion of one- and two-body terms plus two four-body operators ($P_{42}$ and $P_{46}$) to reproduce the 88 reported bending band origin values [9] with an $rms = 0.79\,\mathrm{cm}^{-1}$ ($N = 150$).

We have carried out fits, using the four-body Hamiltonian (13), to the available data for nonrigid bending vibrational modes for $CH_3NCO$, $^{37}ClCNO$, and OCCCO. The molecules $CH_3NCO$ and $^{37}ClCNO$ were previously studied using the 2DVM, but including only interactions up to 2-body in the Hamiltonian [31]. In all cases, the one- and two-body interactions have been included. In the $CH_3NCO$ case we have added an additional (four-body) term, in the $^{37}ClCNO$ case two additional (three-body and four-body) terms, and one extra (four-body) term in the OCCCO case.

We show in Tab. 1 the optimized parameter values for the different molecules under study. For the sake of completeness we include the parameter values for the six molecular species. In all cases $P_{ij}$ coefficients are reported in $\mathrm{cm}^{-1}$ units and we include the total number of bosons $N$, the achieved root mean square $rms$ deviation ($\mathrm{cm}^{-1}$) and the total number of experimental data $N_{data}$ used in the fit. A detailed description of the fitting procedure can be found in Ref. [33]. We include tables with the reported bending band origins, our calculation and state assignment as well as the fit residuals in App. B.

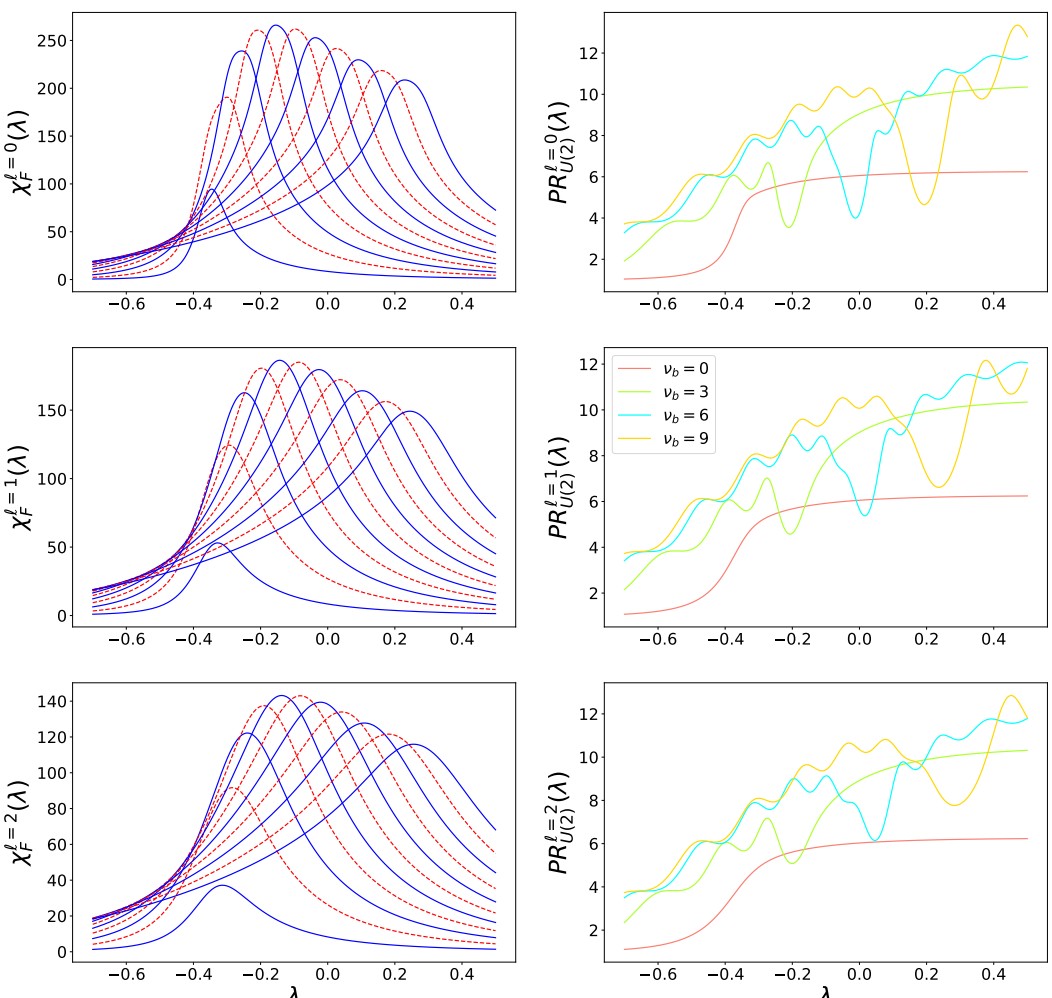

Figure 4: All panels: Results for the optimized eigenstates of the bending degree of freedom of $Si_2C$. The upper, middle, and lower rows show results for $\ell = 0, 1, 2$, respectively. Left panels: QFS for states with $\nu_b = 0, 1, \ldots, 10$ as a function of the $\lambda$ control parameter. Full blue lines alternate with dashed red lines for an easier distinction between adjacent states. Right panels: Participation ratio versus the $\lambda$ control parameter for selected $Si_2C$ bending states ($\nu_b = 0, 3, 6$, and $9$).

| | $CH_3NCO$ | $^{37}ClCNO$ | OCCCO | $HNC^a$ | $Si_2C^a$ | $NCNCS^a$ |
|---|---|---|---|---|---|---|
| $P_{11}$ | 449.5(13) | 760.88(16) | 263.99(15) | 1414.0(4) | 63.8(5) | 331.97(8) |
| $P_{21}$ | -5.477(22) | -7.9142(24) | -2.3308(25) | -29.837(15) | -0.108(18) | -2.0954(6) |
| $P_{22}$ | 7.85(7) | 3.818(14) | 1.300(17) | 15.81(10) | 0.98(5) | 1.190(8) |
| $P_{23}$ | -1.628(4) | -2.1276(6) | -0.6768(4) | -8.054(3) | -0.8117(17) | -0.58578(17) |
| $P_{32}$ | - | - | - | $4.9(10) \times 10^{-2}$ | - | - |
| $P_{33}$ | - | $1.8(7)\times 10^{-5}$ | - | - | - | - |
| $P_{42}$ | - | - | - | - | - | $-2.65(20)\times 10^{-5}$ |
| $P_{43}$ | - | - | $7.2(12)\times 10^{-4}$ | - | - | - |
| $P_{45}$ | $-1.25(23)\times 10^{-5}$ | $-6.61(15)\times 10^{-6}$ | - | - | - | - |
| $P_{46}$ | - | - | - | - | - | $3.48(8)\times 10^{-7}$ |
| $N$ | 78 | 92 | 100 | 40 | 49 | 150 |
| $rms$ | 1.10 | 0.12 | 0.60 | 0.08 | 1.48 | 0.79 |
| $N_{data}$ | 19 | 33 | 36 | 19 | 37 | 88 |

Table 1: Optimized Hamiltonian parameters ($P_{ij}$, in cm$^{-1}$ units) for the selected bending degree of freedom of the molecules under study. Values are provided together with their associated uncertainty in parentheses in units of the last quoted digits. The total vibron number, $N$, the obtained $rms$ of the fit, and the number of reported bending band origins considered in the fit are also included.

$^a$ Fits from a previous work [33].

The CNC bending of $CH_3NCO$ (normal mode $\nu_8$) has a nonrigid character and we have carried out a fit making use of the four-body Hamiltonian (13) to the 19 available experimental data [119], with $\nu_b$ up to 3, and a maximum value of the vibrational angular momentum $\ell = 7$. The parameter resulting from the fit can be found in the first column of Tab. 1. The obtained results, with an $rms = 1.10$ cm$^{-1}$, are rather close to the results previously obtained with the 2DVM including only one- and two-body interactions in the Hamiltonian ($rms = 1.34$ cm$^{-1}$) [31]. We have kept constant the total number of bosons $N$ used in Ref. [31] and there is only a four-body parameter from Hamiltonian (13), $P_{45}$, that significantly improves the quality of the fit. This can be explained due to the complexity of the $CH_3NCO$ spectrum, with two coupled vibrational modes of large amplitude: an internal methyl rotor, with a low energy potential barrier (at approx. 20 cm$^{-1}$), and the CNC bending mode, characterized by a large anharmonicity. This molecule is currently the target of some studies in our group, trying to simultaneously treat the large amplitude bending and the internal rotation within a common algebraic formalism.

We have also performed a fit to the 33 available experimental data for the ClCN bending ($\nu_5$ normal mode) of the $^{37}Cl$ isotopologue of ClCNO [120]. The data set comprises states with bending excitation $\nu_b$ up to 3 and vibrational angular momentum $\ell$ up to 9 units. As in the $CH_3NCO$ case, the bending spectrum of this normal mode has been previously analyzed using the one- and two-body Hamiltonian of the 2DVM, obtaining an $rms$ of 0.71cm$^{-1}$ [31]. Our fit includes two higher-order interactions: $P_{33}$ and $P_{45}$, which allows for a a reduction of the $rms$ to 0.12 cm$^{-1}$. In this case, the interactions introduced are diagonal in neither the $U(2)$ nor the $SO(3)$ basis. As in the previous case, we have used the same total number of vibrons, $N$, than Larese *et al.* [31].

The third molecular species whose bending spectrum has been modeled for its inclusion in the present work is OCCCO. In this case we focus on the CCC bending (normal mode $\nu_7$) and we have carried out a fit to the 36 available experimental term values, with a maximum $\nu_b = 3$ and a maximum $\ell = 12$ [121]. We have included, in addition to one- and two-body

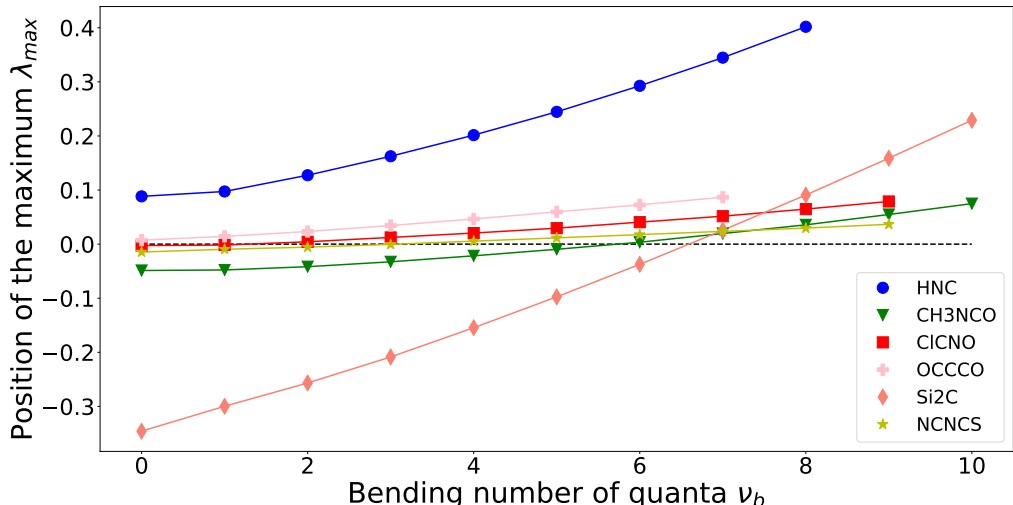

Figure 5: Position $\lambda$ where the QFS takes its maximum value for each state with $\ell = 0$ of selected bending degrees of freedom for HNC (blue), $CH_3NCO$ (dark-green), $^{37}ClCNO$ (red), OCCCO (pink), $Si_2C$ (coral) and NCNCS (olive), versus the bending quantum number $\nu_b$. The dashed black horizontal line marks the $\lambda = 0$ value.

operators, the $P_{43}$ parameter interaction, obtaining an $rms$ equal to $0.60\,\mathrm{cm}^{-1}$. The total number of vibrons has been manually adjusted to $N = 100$.

In summary, we have located where the maximum QFS occurs for states with $\ell = 0$. In all cases, the optimized Hamiltonian can be recovered for $\lambda = 0$. For this reason, attending to the explanation given in section 3, a state with a maximum QFS at a negative $\lambda$ value is located in the $SO(3)$ phase of the ESQPT and should have a bent character. In case the maximum QFS occurs at a positive $\lambda$, the state has a linear character and belongs to the $U(2)$ ESQPT phase.

The Fig. 5 shows the $\lambda$ values at which maxima occur for the states with $\ell = 0$ in all cases examined, including $Si_2C$. HNC and $Si_2C$ can be considered as two textbook examples of a linear molecule and of a nonrigid molecule. As expected, all $\lambda$ values are positive in the HNC case (blue circles). The varying slope between the ground and the first excited states and the rest could be explained by the change from a positive to a negative anharmonicity that characterizes the bending of this molecular species. On the other hand, in the $Si_2C$ case (coral diamonds), the ground state, the fundamental, and the first five overtones have bent character (negative $\lambda$), whereas the rest are linear.

The results for $CH_3NCO$ (green triangles), $^{37}ClCNO$ (red squares), and NCNCS (olive stars) confirm that these molecules are also nonrigid, and their excited states lie closer to the separatrix line between the ESQPT phases than in the $Si_2C$ case. Therefore, their QFS maxima occur in the vicinity of $\lambda = 0$. The states with QFS maximum at negative $\lambda$ values are the ground state plus five excited states for $CH_3NCO$ and the ground state plus two excited states for NCNCS. Therefore, the following excited state would be just above the barrier to linearity. In the $^{37}ClCNO$ isotopologue case, already the bending fundamental has

a maximum at a positive $\lambda$ value and, therefore, only the ground state with $K_a = J = 0$ would have bent character. The last molecule we have decided to include in this work is OCCCO (pink crosses), with all states above a very low energy barrier to linearity, including the ground state.

We have included in the left column of Fig. 7 of App. B the dependence of the QFS with $\lambda$ for the above mentioned five molecules, from where the position of the maxima reported in Fig. 5 have been extracted. In the right column panels of the same figure, we depict the PR in the two bases considered as well as the $\lambda = 0$ QFS values.

## 5 Concluding remarks

In summary, we have introduced a new perspective into excited state quantum phase transitions making use of the quantum fidelity susceptibility in the study of the excited states of the 2D limit of the vibron model. The QFS, a quantity of first importance in Quantum Information Theory, has been chiefly used to characterize ground state quantum phase transitions in different many-body quantum systems. Using results for a 2DVM model Hamiltonian, we have shown how the extension of the QFS from the ground state to encompass excited states provides a convenient tool for the study and characterization of ESQPTs and allows for a fully basis-independent assignment of overtones to one of the possible ESQPT phases in molecular bending spectra. In this regard, these findings nicely complements the information about the ESQPT provided by the PR [79–81], though QFS achieves an unambiguous assignment of states to ESQPT phases even for situations very far from the dynamical symmetry limits.

As an application, we have carried out calculations using a four-body algebraic 2DVM Hamiltonian (13) and fitting bending data from six molecular species: $Si_2C$, HNC, $CH_3NCO$, $^{37}ClCNO$, OCCCO, and NCNCS. The fits to reported band origins of three of them ($Si_2C$, HNC, and NCNCS) have been recently published [33], while the fits for the other three ($CH_3NCO$, $^{37}ClCNO$, OCCCO) have not been previously reported. In all cases, the vibrational bending mode under study is anharmonic, and all but HNC can be considered as nonrigid molecular species, with a feature-rich and complex bending spectrum. A very satisfactory agreement with the reported data has been achieved; the obtained energies and eigenfunctions have been used for the calculation of QFS for the six molecular species.

We have presented a detailed account of the QFS results obtained in the $Si_2C$ case, and an outline of the results for the rest of the molecules. The obtained results provide a satisfactory estimation of the height of the barrier to linearity (which coincides with the ESQPT critical energy) in all cases and the QFS has proved to be a very sensitive tool for the classification of eigenstates as having a linear or bent character.

We are currently working on the study of universality and scaling laws of the QFS in ground state and excited state QPTs for the vibron model and its limits in 1D and 3D as well as in ESQPTs for other quantum systems.

## Acknowledgements

We thank useful discussion with Profs. Lea F. Santos, José M. Arias, and Pedro Pérez-Fernández. Computing resources supporting this work were provided by the CEAFMC and

Universidad de Huelva High Performance Computer (HPC@UHU) located in the Campus Universitario el Carmen and funded by FEDER/MINECO project UNHU-15CE-2848.

**Funding information** This project has received funding from the European Union's Horizon 2020 research and innovation program under the Marie Skłodowska-Curie grant agreement No 872081 and from the Spanish National Research, Development, and Innovation plan (RDI plan) under the project PID2019-104002GB-C21 (JKR, MC, and FPB) and COOPB20364 (MC). This work has also been partially supported by the Consejería de Conocimiento, Investigación y Universidad, Junta de Andalucía and European Regional Development Fund (ERDF), refs. SOMM17/6105/UGR (MC and FPB) and UHU-1262561 (JKR and FPB).

## A    Centrifugal barrier effects

We illustrate the centrifugal barrier effects over QFS and PR with the data calculated for the $Si_2C$ molecule in Fig. 6. We show the QFS (full lines and left ordinate axes) and the PR in the $U(2)$ basis (dashed lines and right ordinate axes) for vibrational angular momentum values $\ell = 0$ (red, first panel), 1 (blue, second panel), 2 (green, third panel), and 3 (orange, fourth panel). These two quantities are excellent probes to look for ESQPT precursors in the 2DVM and other systems. In the panels of this figure, we can appreciate how the bent-to-linear ESQPT precursors weaken for increasing vibrational angular momentum values.

It is known that the ESQPT critical state in bent-to-linear transitions modeled with the 2DVM has a large component in the $\left|n^\ell\right\rangle = \left|\ell^\ell\right\rangle$ element when expressed in the $U(2)$ basis [79–81], which translates into a minimum value of the Participation Ratio. The localization in the U(2) basis becomes softer for higher values of $\ell$. This well known fact can be explained considering the influence of the centrifugal barrier, which hinders the non-zero angular momentum wave function access to the bent-to-linear barrier maximum.

It can be appreciated in the figure how QFS values $\chi_F\left(\lambda = 0\right)$ in the transition state $\nu_b = 6$ diminish as the vibrational angular momentum $\ell$ increases.

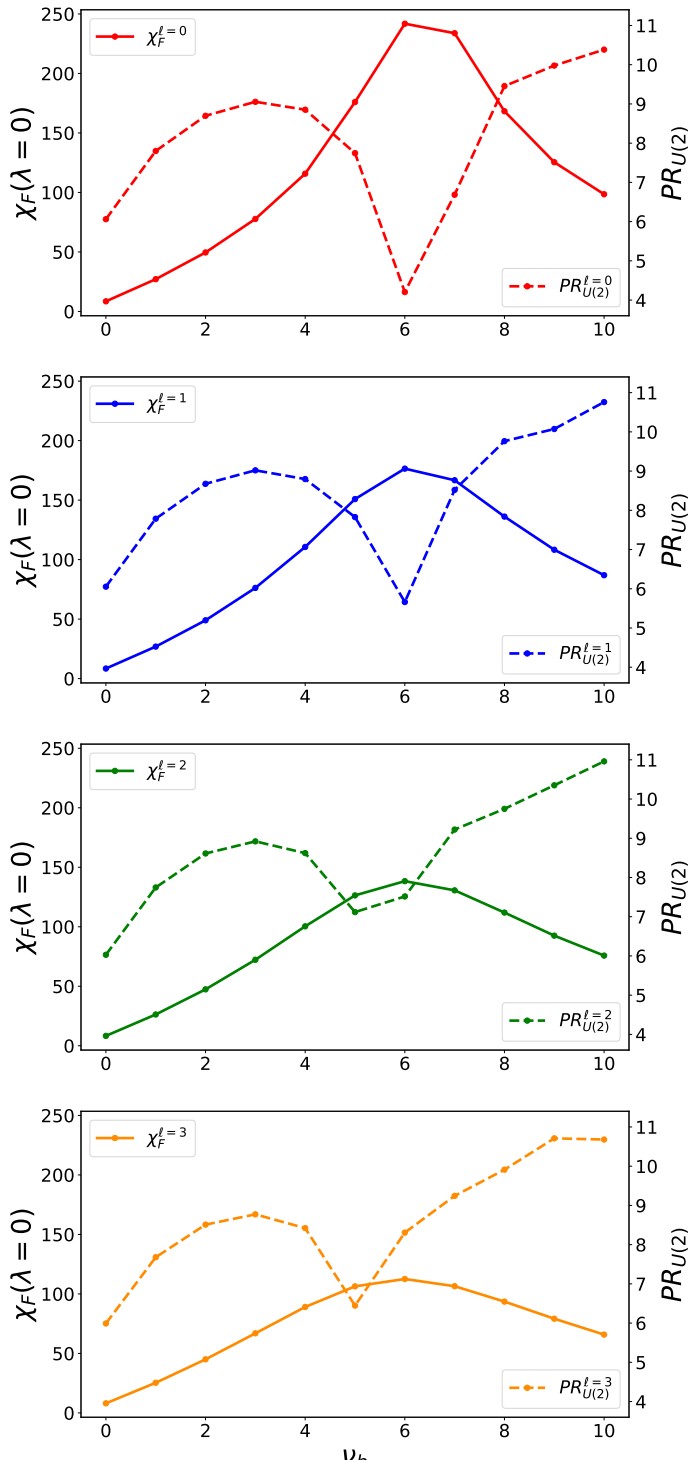

Figure 6:  Si$_2$C bending eigenstates QFS (solid line) and PR in the $U(2)$ basis (dashed line) evaluated for $\lambda = 0$ versus the bending quantum number $\nu_b$. Red, blue, green, and orange lines correspond to $\ell = 0$, 1, 2, and 3, respectively.

# B   Energy fits, residuals, and QFS and PR results

As already mentioned, the procedure followed to fit the Hamiltonian (13) to the available bending origin bands is the one already described in Ref. [33]. For the sake of completeness, we include in this appendix tables including experimental and computed bending band origin values, as well as the resulting residuals. We provide the results for the three species whose fit has not been published yet: $CH_3NCO$ (Tab. 2), $^{37}ClCNO$ (Tab. 3), and OCCCO (Tab. 4). The states are labeled in all cases but the OCCCO one using the bent molecule notation. In the OCCCO case we use the linear molecule quantum labels transforming them to $\nu_b$ values in the figures [33]. The interested reader can find results for HNC, $Si_2C$, or NCNCS in Ref. [33].

We also provide in this appendix the intermediate results needed to reproduce Fig. 5. In the left column of Fig. 7, the QFS for the first bending states is depicted as a function of the $\lambda$ parameter. QFS values for $\nu_b = 0, 2, \ldots$ states are depicted with full blue lines, while for $\nu_b = 1, 3, \ldots$, the QFS is depicted with dashed red lines. The $\lambda$ parameter values corresponding to the maximum QFS value for each state in these panels are the ones depicted in Fig. 5.

The panels in the right column of Fig. 7 display the $\lambda = 0$ QFS ($\chi_F(\lambda = 0)$, red full lines), and PR in the U(2) (blue dashed lines) and SO(3) (green dashed lines) bases as a function of the number of quanta of bending excitation for each molecule. The QFS shares axes and ticks with the corresponding plot on the left column, whereas the scale for the PR is located on the right side of the right column panels.

| $\nu_b$, $\ell$ | Exp. E. | Calc. E. | Exp.-Calc. | $\nu_b$, $\ell$ | Exp. E. | Calc. E. | Exp.-Calc. |
|---|---|---|---|---|---|---|---|
| 0, 0 | 0.0 | 0.0000 | 0.0000 | 0, 3 | 80.0 | 78.8075 | 1.1925 |
| 1, 0 | 182.2 | 183.5477 | -1.3478 | 1, 3 | 268.6 | 268.5326 | 0.0674 |
| 2, 0 | 357.9 | 358.8119 | -0.9119 | 2, 3 | 454.0 | 453.1013 | 0.8987 |
| 3, 0 | 525.1 | 523.8246 | 1.2754 | 0, 4 | 140.6 | 139.5489 | 1.0511 |
| 0, 1 | 8.4 | 8.7997 | -0.3997 | 1, 4 | 333.4 | 333.3653 | 0.0347 |
| 1, 1 | 191.4 | 193.0995 | -1.6995 | 0, 5 | 217.5 | 217.0241 | 0.4759 |
| 2, 1 | 368.6 | 369.6086 | -1.0086 | 1, 5 | 415.5 | 415.4570 | 0.0430 |
| 0, 2 | 36.8 | 35.1316 | 1.6684 | 0, 6 | 311.1 | 310.8790 | 0.2210 |
| 1, 2 | 222.3 | 221.5793 | 0.7207 | 1, 6 | 513.4 | 514.2226 | -0.8226 |
| 2, 2 | 402.1 | 401.4326 | 0.6674 | 0, 7 | 420.0 | 420.7589 | -0.7589 |

Table 2: Experimental [119] and calculated band origins and residuals for the CNC bending mode of $CH_3NCO$. Units of $cm^{-1}$.

| $\nu_b$, $\ell$ | Exp. E. | Calc. E. | Exp.-Calc. |
|---|---|---|---|
| 0, 0 | 0.0 | 0.0000 | 0.0000 |
| 1, 0 | 120.9 | 120.8932 | 0.0068 |
| 2, 0 | 258.5 | 258.6117 | -0.1117 |
| 3, 0 | 432.0 | 432.1241 | -0.1241 |
| 0, 1 | 17.5 | 17.6438 | -0.1438 |
| 1, 1 | 167.9 | 167.7928 | 0.1072 |
| 2, 1 | 335.1 | 335.0671 | 0.0329 |
| 3, 1 | 525.3 | 525.4485 | -0.1485 |
| 0, 2 | 55.6 | 55.7618 | -0.1618 |
| 1, 2 | 227.8 | 227.7131 | 0.0869 |
| 2, 2 | 415.1 | 415.1037 | -0.0037 |
| 3, 2 | 620.1 | 620.1478 | -0.0478 |
| 0, 3 | 108.1 | 108.2590 | -0.1590 |
| 1, 3 | 297.6 | 297.4773 | 0.1227 |
| 2, 3 | 500.5 | 500.4730 | 0.0270 |
| 3, 3 | 717.9 | 717.9603 | -0.0603 |
| 0, 4 | 171.8 | 171.9164 | -0.1165 |

| $\nu_b$, $\ell$ | Exp. E. | Calc. E. | Exp.-Calc. |
|---|---|---|---|
| 1, 4 | 375.5 | 375.4203 | 0.0797 |
| 2, 4 | 591.3 | 591.2717 | 0.0283 |
| 3, 4 | 819.6 | 819.4564 | 0.1436 |
| 0, 5 | 244.7 | 244.7522 | -0.0522 |
| 1, 5 | 460.5 | 460.4442 | 0.0558 |
| 2, 5 | 687.2 | 687.2587 | -0.0588 |
| 3, 5 | 925.0 | 924.7723 | 0.2277 |
| 0, 6 | 325.4 | 325.4264 | -0.0264 |
| 1, 6 | 551.8 | 551.7516 | 0.0484 |
| 2, 6 | 788.1 | 788.1261 | -0.0261 |
| 0, 7 | 413.1 | 412.9750 | 0.1250 |
| 1, 7 | 648.7 | 648.7310 | -0.0310 |
| 2, 7 | 893.5 | 893.5698 | -0.0698 |
| 0, 8 | 506.8 | 506.6723 | 0.1277 |
| 1, 8 | 750.8 | 750.8960 | -0.0960 |
| 0, 9 | 606.1 | 605.9530 | 0.1470 |
| 1, 9 | 857.6 | 857.8490 | -0.2490 |

Table 3: Experimental [120] and calculated band origins and residuals for the ClCN bending mode of $^{37}$ClCNO. Units of cm$^{-1}$.

| $n^\ell$ | Exp. E. | Calc. E. | Exp.-Calc. |
|---|---|---|---|
| $2^0$ | 60.70 | 60.2640 | 0.4360 |
| $4^0$ | 144.30 | 144.3234 | -0.0234 |
| $6^0$ | 244.70 | 244.4930 | 0.2070 |
| $1^1$ | 18.26 | 18.6991 | -0.4391 |
| $3^1$ | 97.22 | 97.2062 | 0.0138 |
| $5^1$ | 191.06 | 191.4044 | -0.3444 |
| $7^1$ | 299.26 | 298.0710 | 1.1890 |
| $2^2$ | 46.11 | 46.5075 | -0.3975 |
| $4^2$ | 137.26 | 137.4453 | -0.1853 |
| $6^2$ | 239.57 | 240.1829 | -0.6129 |
| $8^2$ | 352.91 | 352.8677 | 0.0423 |
| $3^3$ | 80.62 | 80.7559 | -0.1359 |
| $5^3$ | 181.02 | 181.2481 | -0.2281 |
| $7^3$ | 290.52 | 291.1644 | -0.6444 |
| $9^3$ | 407.97 | 409.2399 | -1.2699 |
| $4^4$ | 120.37 | 120.1674 | 0.2026 |
| $6^4$ | 228.23 | 228.4397 | -0.2097 |
| $8^4$ | 345.27 | 344.4927 | 0.7773 |

| $n^\ell$ | Exp. E. | Calc. E. | Exp.-Calc. |
|---|---|---|---|
| $10^4$ | 466.79 | 467.3681 | -0.5781 |
| $5^5$ | 164.49 | 163.9965 | 0.4935 |
| $7^5$ | 278.61 | 278.8217 | -0.2117 |
| $9^5$ | 401.59 | 400.1995 | 1.3905 |
| $11^5$ | 528.08 | 527.3543 | 0.7257 |
| $6^6$ | 212.39 | 211.7692 | 0.6208 |
| $8^6$ | 331.89 | 332.2356 | -0.3456 |
| $10^6$ | 458.01 | 458.2881 | -0.2781 |
| $7^7$ | 263.65 | 263.1749 | 0.4752 |
| $9^7$ | 388.95 | 388.5714 | 0.3786 |
| $11^7$ | 518.15 | 518.7640 | -0.6140 |
| $8^8$ | 317.94 | 318.0124 | -0.0724 |
| $10^8$ | 447.91 | 447.7643 | 0.1457 |
| $9^9$ | 375.65 | 376.1605 | -0.5105 |
| $11^9$ | 510.04 | 509.7902 | 0.2498 |
| $10^{10}$ | 436.77 | 437.5598 | -0.7898 |
| $11^{11}$ | 501.91 | 502.2009 | -0.2909 |
| $12^{12}$ | 570.68 | 570.1168 | 0.5632 |

Table 4: Experimental [121] and calculated term values and residuals for the CCC bending mode of OCCCO. Units of cm$^{-1}$.

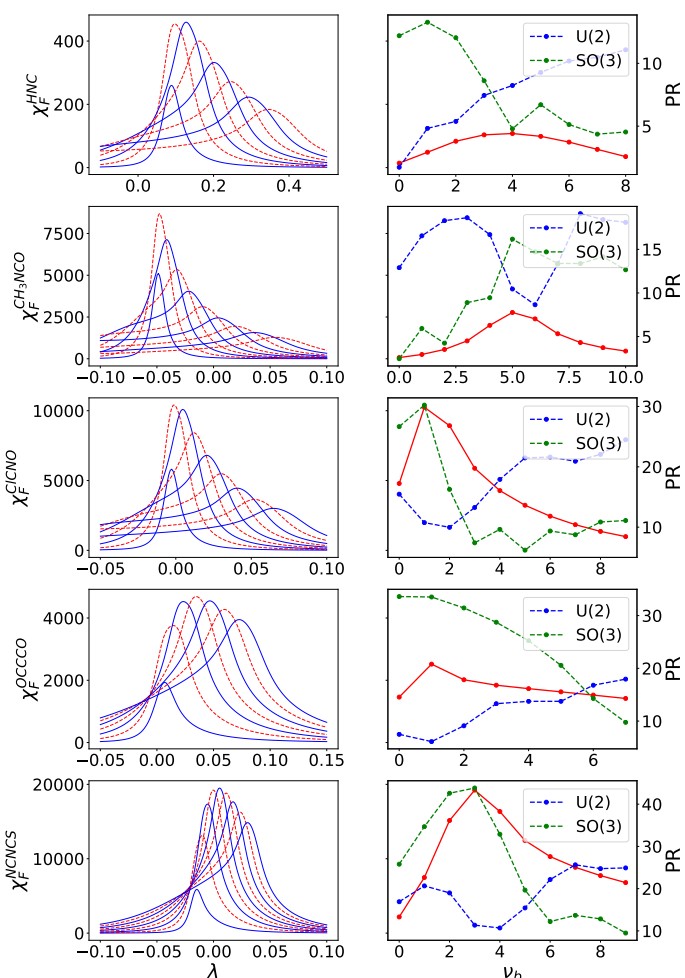

Figure 7: Left column: QFS for states with $\ell = 0$, $\chi_F(\lambda)$, versus the control parameter, $\lambda$. Right column: QFS for $\lambda = 0$ (solid red line) using the same scales as in the left panels and PR in the $U(2)$ (blue dashed line) and $SO(3)$ (green dashed line) bases (right axes scale). Results for the five molecules that have been selected to illustrate the QFS results in 2DVM systems, from top to bottom: HNC, $CH_3NCO$, $^{37}ClCNO$, OCCCO, and NCNCS.

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
