# Peer review of "Quantum fidelity susceptibility in excited state quantum phase transitions: application to the bending spectra of nonrigid molecules"

_SciPost Physics_

## Round 2 · Referee Report · Anonymous (Referee 4) · 2021-5-17

Report

The work “Quantum fidelity susceptibility in excited state quantum phase transitions: application to the bending spectra of nonrigid molecules” by Khalouf-Rivera, Carvajal and P erez-Bernal represents an extension of the analysis performed by the same authors in Ref.[33]. The purpose of these works is to identify (ES)QPTs and quantum phases in the spectrum of the two-dimensional vibron model with a general Hamiltonian containing terms up to the four-body interactions. As shown in the previous work, the model in this form provides a satisfactory description of experimental data on numerous molecules, so it is certainly worth of attention.

In the previous work, the main tool to distinguish quantum phases was the participation ratio, which measures the localization of eigenstates in a selected basis. However, this quantity fails to perform the desired task of an ESQPT indicator in generic systems in which ESQPTs do not show localization effects in the initial eigenbasis. Here the authors employ the so-called fidelity susceptibility (FS) of the eigenstates. This is just the lowest-order (quadratic) term of the fidelity expansion in powers of the local parameter change. Since the FS quantifies how rapid is the local change of the selected eigenstate with the control parameter, it can serve as an indicator of structural changes and ESQPTs in the spectrum.

The work is well written and presents interesting new results. The authors show the behavior of the FS not only in the l=0 eigenstates, that undergo (in the infinite size limit) the ESQPT, but also in the l>0 ones, which have no ESQPT and show only fading signatures of the structural change. The method based on the FS is proven to be a useful heuristic tool for fast localization of rapid structural changes of eigenstates in the model parameter space, including excited-state critical effects. Applications of this type may follow also in other models, whose complexity hinders the full semiclassical analysis needed for the identification of ESQPTs.

I recommend the present work for publication in the SciPost Physics. Below I give a few minor comments which may help to improve the clarity of the paper.

Requested changes

Here are some reccommendations for the authors:

1) I believe that the authors should mention that the FS is supposed to be large not only in crossing the QPT and ESQPT critical points, but also near any sufficiently sharp avoided crossing of levels. This directly follows from Eq.(8). So particularly in chaotic systems the present approach may face a trouble.

2) Concerning the origin of the concept of quantum monodromy, it would be convenient to cite besides the Child’s work also Cushman and Duistermaat, Bulletin of Am. Math. Soc. 19, 1988.

3) On page 3 the authors say that “For a system with n effective degrees of freedom, the order of the derivative of the level density that is non-analytic is n–1.” However, this is true only in some cases (though the most common ones), particularly for ESQPTs caused by nondegenerate stationary points. A short comment on the other cases would be helpful.

4) Equation (7) deserves more motivation. It would be worth to say that the first derivative of the fidelity vanishes, so that the FS is the leading-order term in the fidelity expansion.

5) The introduction of two control parameters, namely xi and lambda, is somewhat confusing, especially in Eq.(12), where both parameters play essentially the same role. Maybe a few motivating words would be in order here.

  • validity: high
  • significance: good
  • originality: good
  • clarity: high
  • formatting: excellent
  • grammar: perfect

Author:  Francisco Perez-Bernal  on 2021-06-08  [id 1494]

(in reply to Report 1 on 2021-05-17)

We are very grateful for the careful reading of the manuscript by the invited referee and for his very positive opinion on our work. As she/he has stated, the fidelity susceptibility is very sensitive to eigenstate changes and this fact makes this quantity a fine probe to locate eigenstates with respect to an ESQPT separatrix, even for finite system sizes and angular momentum values other than zero. We are currently planning to extend the present approach to other systems and models, exploring a novel -and convenient- way to assign excited states to ESQPT "phases."

We proceed to answer the referee's five recommended changes:

(1) We thank the referee for this insightful remark. We have changed the text under Eq. (11) in page 9 to explain this point and the sensitivity of the QFS to avoided energy crossings.

(2) The reference has been included.

(3) We thank the referee for this suggestion that corrects our statement. The sentence has been recast to exclude degenerate stationary point cases.

(4) We agree with the referee that the QFS needs to be properly introduced in the paper, especially for those non-familiar with this quantity. We have introduced a sentence stating that the QFS is the leading term in the series expansion of the fidelity.

(5) We have introduced an explanatory sentence and changed \xi to its numerical value to clarify the role of the control parameter \lambda.

The final version with the changes described above will be uploaded once we include possible changes suggested by the expected second referee report.

---

## Round 2 · Referee Report · Anonymous (Referee 3) · 2021-7-6

Strengths

1. The paper is well written and ordered.
2. The idea of using the quantum fidelity susceptibility (QFS) and comparing it to the use of the participation ratio to characterize the excited state phase transitions is a good one.

Weaknesses

1. A large fraction of this paper has been published before, by the same authors! See their article in the Journal of Quantitative Spectroscopy & Radiative Transfer [261 (2021) 107436].
2. The lengthy discussion of the 2D vibron model is nearly identical in the two papers.
3. The general Hamiltonian given in Eq. 13 has been discussed in the JQSRT article, and the *same* fit results (Table 1) were presented there. There is no need to reproduce them here.
n. The addition of QFS to the author's narrative does not warrant an article with the size and scope of this manuscript.

Report

This has been a very difficult paper to review because it is too similar to the author's previous article in JQRST. I spent a lot of time trying to find the differences between the article. Those differences, related to the use of quantum fidelity susceptibility to locate the control parameters corresponding to excited state phase transitions, contain the nucleus of an interesting paper, but only if they are presented differently, without excessive repetition of previously published work.

Requested changes

I can only make one requested change: focus your paper far more sharply than you have done! Avoid republishing your previous discussions and results and instead focus on the only relevant idea for this manuscript: the quantum fidelity susceptibility. I think that the entire manuscript has to be rewritten, referring to previously published work instead of reproducing it. This will clarify the value of using the QFS instead of burying it in discussion and analysis given elsewhere.

  • validity: good
  • significance: good
  • originality: good
  • clarity: high
  • formatting: good
  • grammar: good

Author:  Francisco Perez-Bernal  on 2021-07-19  [id 1583]

(in reply to Report 2 on 2021-07-06)
Category:
reply to objection

We thank very much the second referee for her/his careful reading of our manuscript and for her/his suggestions to improve it. We are glad that the referee finds the submitted manuscript well-written and that the use of Quantum Fidelity Susceptibility (QFS) to characterize ESQPTs is a good idea.

As regards the weaknesses pointed out in the referee report, the referee expresses her/his concern about the repetition of results and the overlap with a previous paper of us (Journal of Quantitative Spectroscopy & Radiative Transfer [261 (2021) 107436]). We agree that this overlap exists and our aim in writing the paper in this way was to make it as self-contained as possible and more accessible to a reader not familiar with the 2DVM. In our opinion, though there was an evident overlap between the submitted work and our previous article, we tried -maybe not very successfully considering the referee
recomendation- to make a clear distinction between what has been previously published and the new results. However, we understand her/his concern on this point and we have carried out several changes following the referee suggestion. The new version of the manuscript avoids the repetition of previously published results, making clear the purpose of this work, and leaving no place for confusion between this work and any of our previous manuscripts.

Summary of changes:

#1 Abstract: We have rewritten it to make clear that we out of the six molecular species whose bending modes where studied, only the fits for three of them have been performed with this work in mind, the fit for the other three being recently published in the above mentioned JQSRT paper.

#2 Page 4. Last paragraph of the introductory section. we replaced "In Sect. 4, we introduce the four-body Hamiltonian used to fit to experimental data and we apply the formalism to the Si2C molecule, a well-known example of nonrigid molecule [6]." by "In Sect. 4, we apply the formalism to the Si2C molecule, a well-known example of nonrigid molecule [6].".

#3 Page 5. We removed the branching rules between quantum labels in Chains I and II -old version Eq. (2)- replacing them by several references for the interested reader.

#4 Page 6. We removed Eq. (4) in the old version -model hamiltonian matrix elements in Chain I basis- and replace it by the reference to [29].

#5 Page 6. We removed Figure 1 and the text addressing this figure, as the correlation energy diagram for the model Hamiltonian can be found elsewhere and, though it may be enlightening to the reader nonfamiliar with the 2DVM, it can be found elsewhere.

#6 Page 6. We introduced at the end of section 1 the participation ratio as a tool to study wave function localization in ESQPTs, instead of making it in section 2 as in the original manuscript version. In this way, we concentrate in section 2 in the QFS which is the truly important contribution of the present work.

#7 Page 7. Section 3 is now entirely devoted to the QFS.

#8 Page 9. We removed the last sentence in the first paragraph of Section 4: "In particular, we show ... OCCCO." and we have recasted the text following Eq. (11) to make clear that the matrix elements of the four-body Hamiltonian can be found in the JQSRT reference and that for three of the addressed cases (Si2C, NCNCS, and HNC) we use eigenstates from the fits published in JQSRT, making clear that Table 1 only contains unpublished fit results.

#9 Page 13. We have removed any discussion of priorly published fit results, replacing them by the corresponding Ref. to the JQSRT article. E.g. First paragraph of Subsection 4.1 and second and third paragraphs of Subsection 4.2.

#10 Page 15, Table 1 in the new version only includes fit to bending degrees of freedom that have not been previously published (molecular species CH3NCO, 37ClCNO, and OCCCO. The references to the table in the text have been changed accordingly.

#11 Page 17. The beginning of the last section (Section 5) has been rewritten to emphasize that the main purpose of this manuscript is the use of QFS in the study of ESQPTs. The second paragraph has also been rewritten to make clear what is new in this article and what stems from the previously published JQSRT paper.

We hope that with this changes the article is now completely acceptable for the referee and we cleared any doubts the overlap of this work with our previous JQSRT paper could have raised.

---

## Editorial Decision

resubmitted